# Four Minutes of Sprint Interval Training Had No Acute Effect on Improving Alertness, Mood, and Memory of Female Primary School Children and Secondary School Adolescents: A Randomized Controlled Trial

**DOI:** 10.3390/jfmk5040092

**Published:** 2020-12-14

**Authors:** Terence Chua, Abdul Rashid Aziz, Michael Chia

**Affiliations:** 1Physical Education and Sport Science Academic Group, National Institute of Education, Nanyang Technological University, Singapore 637616, Singapore; terence.chua@nie.edu.sg; 2Sport Medicine and Sport Science, Singapore Sport Institute, Singapore 397630, Singapore; abdul_rashid_aziz@sport.gov.sg

**Keywords:** sprint-interval training, learning in youths, acute exercise, school, attention

## Abstract

We investigated whether a 4-min sprint interval training (SIT) protocol had an acute effect (15 min after) on improving alertness, mood, and memory recall in female students. Sixty-three children and 131 adolescents were randomly assigned to either a SIT or control (CON) group by the class Physical Education (PE) teachers. The SIT intervention was delivered twice a week for 3 weeks. SIT participants performed three, 20-s ‘all-out’ effort sprints interspersed with 60-s intervals of walking while CON group sat down and rested. PE lessons were arranged such that the first two sessions were to familiarise participants with the SIT protocol leading to acute assessments conducted on the third session. On that occasion, both groups rated their alertness and mood on a single-item hedonic scale and underwent an adapted memory recall test. The same assessments were administered to both groups fifteen minutes after delivery of SIT intervention. A 4-min SIT involving three, 20 s ‘all-out’ effort intensity sprints did not have an acute main effect on improving alertness, mood and, memory recall in female children (*η_p_*^2^ = 0.009) and adolescents (*η_p_*^2^ = 0.012). Students’ exercise adherence and feedback from PE teachers are indicatives of the potential scalability of incorporating SIT into PE programmes. Different work-to-rest ratios could be used in future studies.

## 1. Introduction

Emergent literature suggests that a single session of aerobic exercise has beneficial effects on mood [1] and cognition [2] in adults. Reviews and meta-analyses have found that acute and chronic exercise improves attention and memory in children and adolescents [3,4]. These two domains of cognitive functions are essential for learning [5].

Improved cognition and enhanced mood after an acute bout of aerobic exercise may be explained by psychological and neurophysiological mechanisms. Exercise acts as an arousal stimulus [6]. Synthesis of brain-derived neurotropic factors (BDNF) is up-regulated leading to the activation of a pathway that initiates neuroplasticity and neurogenesis of the hippocampus [7,8]. Increased blood circulation during exercise also promotes more oxygen being delivered to the brain [9]. However, these underlying mechanisms are said to be dependent on exercise intensity. A meta-analysis revealed that studies utilising higher intensity exercise reported greater acute cognitive benefits [2]. The intensity of exercise can also affect mood differently [10]. Therefore, higher intensity exercise may be necessary for any potential positive effect on mood and cognition to be maximised. The acute and chronic exercise effects cannot be considered in isolation. It is explained that exercise training both in the short term (immediate and/or soon after) and long term (days, weeks, months, and years) increases the capacity for exercise, thereby permitting more vigorous and/or more prolonged individual exercise sessions and a more significant acute effect [11]. As such, an acute response to an exercise intervention refers to the transient effects of the exercise immediately or soon after the intervention.

Mood, alertness, and memory have facilitative influences on learning. These states of learning are different and but at the same time can be interdependent on each other. Young people experience a range of emotions, from positive emotions like enjoyment to negative emotions like anger, boredom, and anxiety in the process of learning. When negative emotions arise, learning may become a less enjoyable process to them. Consequently, they may be less motivated to interact with teachers and classmates [12,13]. The process of learning involves a continuous effort and awareness to inhibit the shift of attention to irrelevant activities [14,15]. Otherwise, a phenomenon called attentional bottleneck may manifest [16]. Children and adolescents spend six or more hours of their waking time in school, with much of those time spent engaging in sedentary activities due to classroom curriculum requirements and the inactive nature of most post-curriculum school-based activities [17]. Even though a single session of exercise is unlikely to cause impact on mood and cognition lasting throughout the schooling day, it may be opportunistic for qualified physical education teachers to enrich their students’ formal learning experience by harvesting the immediate benefits of exercise.

The impact of interval training on cognition and learning in young people has generated research interest in recent years. Interval training can be characterized as sprint-interval training (SIT) or high-intensity interval training (HIIT). SIT involves very short bouts of ‘all-out’ effort sprints while HIIT involves relatively intense but submaximal workloads corresponding to 80–100% of maximal heart rate. Both forms of interval training are interspersed with periods of lower-intensity recovery [18]. Interval training resembles the sporadic patterns of physical play of free-living young people [19,20] and is generally well-tolerated [21,22]. The attractiveness of interval training is that the time invested to complete the exercise is only a fraction of that in traditional endurance training. This time-saving exercise modality is shown to be effective in improving cardiorespiratory fitness and certain cardiovascular and metabolic disease biomarkers in healthy and overweight youths when implemented as an exercise programme over several weeks to a few months [23,24,25].

Among various interval-training protocols that were studied in the literature, as little as three, 20 s all-out effort cycling sprints performed thrice a week for six weeks has shown to elicit chronic skeletal muscle adaptations linked to increased cardiorespiratory fitness as well as improved cardio-metabolic health biomarkers in adults [26]. This protocol involves a total work duration of 1 min. It is one of the shortest SIT methods to date to demonstrate positive health-related outcomes, albeit in adults. In terms of the acute effect of interval training on cognition in children and adolescents. Previous studies examining the acute effect of interval training on cognition in children and adolescents had used total work durations which were longer than that of Gillen and his colleagues’ protocol. Some researchers used shorter intervals of exercise (10–20 s) while others used longer intervals of exercise (30 s) repeated between 8 and 16 times [27,28,29]. A 15- to 20-s bout of ‘all-out’ intensity sprint is suggested to be more palatable for children and adolescents rather than sprint bouts of longer durations (e.g., 30 s or 40 s) [30].

Interestingly, a recent study showed that embedding interval-training within the school day has led to increased moderate-intensity physical activity levels in adolescents [31]. Girls are oftentimes less active than boys [32,33,34] and hence the use of interval-training to increase physical activity has potential implications on them. Besides, girls are an under-served group compared to boys as research evidence in this area is scarce. Therefore, the primary objective of the study was to investigate the acute effect of a three, 20 s ‘all-out’ effort sprints on self-reported mood and alertness, and memory recall of female children and adolescents when delivered within a physical education (PE) setting. A secondary aim of the study was to gain perspectives from PE teachers about the feasibility of infusing SIT into physical education classes.

## 2. Materials and Methods

### 2.1. Enrolment of Participants and Institutional Ethical Clearance

One independent all-girl primary school and secondary school were invited to take part in the study via convenience sampling. The heads of PE department of both schools were contacts of the principal investigator. The school principal gave her consent and permission for conducting the research in schools was granted by the Ministry of Education. The study protocol and ethics was approved by the Institutional Review Board of the university on 4 April 2018 (IRB-2018-02-009-4). Upon discussion with the PE departments, one Primary 5 and one Secondary 1 cohorts were selected for the study. Written child assent and informed consent from the parents were obtained. Sixty-six female children (9–10 years old) and 131 female adolescents (12–13 years old) from the selected cohorts were enrolled in the study.

### 2.2. Study Design

The research employed a parallel, randomized control study design with a 1:1 allocation ratio. Participants were randomly assigned to either a SIT or control (CON) group at the individual level by the class PE teachers. The SIT intervention commenced twice a week for 3 weeks (6 sessions) and all PE lessons were conducted in the mornings, between 8 and 10 am. Following at least two separate familiarization sessions, the acute effects of SIT was assessed. Of the thirteen classes involved in the study, a few missed 1–2 SIT sessions due to timetabling constraints as reported by the respective PE teachers (i.e., PE lesson was cancelled when it fell on a public holiday).

### 2.3. SIT Protocol and Familiarisation Sessions Leading to the Day of Acute Assessment

The SIT group performed a three-minute warm-up routine (light lower limb stretches) followed by three 20 s ‘all-out’ effort running shuttle-sprints interspersed with intervals of 60 s of walking. The 3 × 20 s sprint bouts adapted from Gillen and his colleagues’ work is one of the shortest SIT protocols to date. A work-to-rest ratio of 1:3 may be appropriate as a study showed that female children and adolescents were able to replicate the peak power generated in the first bout into subsequent bouts of the Wingate Anaerobic Test better than adult women [21]. This suggests that an active rest interval of 60 s derived from the 1:3 ratio may be sufficient for female children and adolescents to re-generate peak anaerobic power. The CON group did not receive the SIT intervention. Instead, they sat down and cheered on the SIT group during the shuttle sprints. Duration of the SIT protocol was four minutes and when warm-up was included, the total exercise session was less than eight minutes.

Prior to the study, the qualified PE teachers involved in the research were trained on how to deliver the SIT intervention and acute assessments by the principal investigator and his research assistant. The first two sessions were delivered to familiarise participants what an ‘all-out’ effort entails in the lead up to the acute assessments conducted on the third session. Each SIT participant was paired with a participant from the CON group who was tasked to count the number of sprint-shuttles completed over a marked out 20-m distance by their partner (data not presented). Upon the teacher’s cue, the SIT group sprinted as hard as they could back and forth the marked-out distance for 20 s. Immediately after the 20 s sprint, participants walked to and from the marked-out distance for 60 s. With 15 s to go, participants were instructed to return to the starting position and get ready for the second bout. The process of a 20-s sprint bout followed by a 60-s walking recovery was repeated. Participants then performed their third and final bout of 20 s sprint and recovered right after for 60 s. Throughout the shuttle-sprints, partners of participants from the SIT group counted aloud the number of sprint shuttles completed and cheered the participants on to match or better the number of sprint shuttles completed in the previous bout. Most participants were able to match the number of shuttles completed in the previous sprint (data not reported). Indirectly, this indicated that participants were able to provide a maximal effort throughout the three sprint bouts.

### 2.4. Acute Assessment of Alertness, Mood, and Memory Recall (before and 15 min after SIT)

The primary outcome measures (i.e., alertness, mood, and memory recall) were assessed during the 3rd PE lesson. Before the SIT intervention was delivered, participants were asked to rate their mood and alertness from 1–10 on a mood scale [35] and a self-constructed alertness scale (i.e., higher number indicated better mood and greater alertness). The questions posed to them were: ‘How are your mood right now? Please circle a number that best represents your current mood.’ and ‘How alert, watchful or attentive are you right now? Please circle a number that best describes you.’ The scale consists of faces with expressions from frown to smiles above the number 1, 5, and 10 in gradations that are intended to reflect a progressive change of feelings. Hedonic scales like these are commonly administered to young children in consumer and food preference research. Although its psychometric properties for assessing mood and alertness are not established, another study has adapted it for the same purpose as the present study [31].

In addition, participants underwent an adapted version of the Rey Auditory Verbal Memory Recall test (RAVRT) [36]. The RAVRT was previously administered to the same age group of participants [37]. Instead of the full five trials, a single trial of recall was used in the present study. The PE teacher read out a list of 15 unrelated nouns each containing two syllables to the participants at a speed of one word per second. After the last word was read out, participants wrote down as many words as they could recall (order and spelling of words were not important). The number of correct words recalled by each participant was recorded. The RAVRT is commonly used in clinical research and practice and has a robust construct validity and internal consistency (Cronbach’s alpha coefficient of 0.8) which was found to be closely associated with other tests of verbal learning which renders it to be a valid and reliable psychometric instrument.

Fifteen minutes after the SIT intervention was delivered, the same acute assessments were administered but in a different order. A different word list was used in the memory recall test of which the words are what students have learnt before. The delay of 11–20 min after exercise was reported to be the window of opportunity for observing the greatest positive effects on cognition. Positive effects may diminish beyond 20 min whereas assessing too soon after exercise may result in negative effects [2]. The study flow from enrolment of schools to data analysis of acute measures is shown in Figure 1 below.

At the end of the study, PE teachers were polled on their perceptions on the feasibility of incorporating SIT-type activities into their PE curriculum (i) to get students fit for sports and (ii) to get students healthy. The PE teachers provided their ratings on a self-constructed 5-point scale, with 1 being ‘I do not find it feasible’ and 5 being ‘I find it very feasible’. To measure overall exercise adherence rate, the average percentage of the number of SIT participants who completed all SIT sessions was divided by the number of SIT participants in each class.

### 2.5. Statistical Analyses

SPSS Version 23 (IBM Corp., Armonk, NY, USA) was the statistical tool used. Normality of data and homogeneity of variance within each group was assessed. Missing data was replaced by series mean of each group. A 2 × 2 repeated-measures analysis of variance (ANOVA) was performed to analyse the main intervention effect. Given there were only two levels of measurements of the outcome variables (i.e., before condition and 15 min after condition), the assumption of sphericity was not violated. The measure of effect size was reported as partial eta square, where *η_p_*^2^ = 0.01–0.05 was interpreted as a small effect size, *η_p_*^2^ = 0.06–0.13 was interpreted as a medium effect size and *η_p_*^2^ = 0.14 or greater was interpreted as a large effect size. The level of statistical significance was determined as *p* < 0.05. Descriptive statistics (mean ± SD) for all outcome variables were reported.

## 3. Results

### 3.1. Acute Changes in Self-Reported Alertness and Mood, and Memory Recall Score

Separate 2 × 2 repeated-measures ANOVAs were performed to compare the acute effect of a 4-min SIT protocol on improving alertness, mood, and memory recall with the CON group. The results of the univariate analysis of each outcome variable is presented in Table 1 for female children and in Table 2 for female adolescents. Multiple Analysis of Variance (MANOVA) revealed that there was no significant main effect of the SIT protocol on improving all three conditions of learning in female children, *F*(3, 59) = 0.168, *p* = 0.918, *η_p_*^2^ = 0.008, as well as in female adolescents, *F*(3, 127) = 0.528, *p* = 0.664, *η_p_*^2^ = 0.012. The time delay of 15 min following SIT has a significant effect on memory recall, with female children performing better (*F*(1, 127) = 5.929, *p* = 0.018, *η_p_*^2^ = 0.089) while female adolescents performed poorer (*F*(1, 59) = 12.801, *p* = 0.001, *η_p_*^2^ = 0.09) in the RAVRT after SIT. The pre-to-post-test change in self-reported alertness and mood, and memory recall scores between SIT and CON groups were not significantly different (*p* > 0.05). These results indicated that a four-minute SIT bout involving a combined one minute of ‘all-out’ effort sprints had no effect on improving alertness, mood, and memory recall in female children and adolescents.

### 3.2. Exercise Adherence and Teachers’ Perceptions on Embedding SIT-Type Activities into the PE Curriculum

Most of the female participants (primary school children: 91.8 ± 5.0%, secondary school adolescents: 93.7 ± 9.2%) completed all SIT sessions that were delivered. Common reasons for their absenteeism reported by the class PE teachers were taking sick leave (not related to the exercise) and being out of school for inter-school competitions on the day of PE lesson. Six PE teachers involved in the study rated 4.2 and 4.0 (out of 5) on their beliefs in the feasibility of infusing SIT-type activities in their PE curriculum to get (i) students fit for sport and (ii) to keep students healthy, respectively. None of the participants were reported to have sustained injuries resulting from the ‘all-out’ intensity sprints under the tutelage of the qualified PE teachers.

## 4. Discussion

The primary objective of the study was to examine the acute effect of a SIT protocol involving three, 20-s ‘all-out’ effort sprints on improving mood, alertness and memory recall in female primary school children and secondary school adolescents. Of interest was also the PE teachers’ perspectives on using SIT in a lesson setting (i.e., teachers’ thoughts on infusing SIT-type activity in PE lessons as an intervention to get students fit for sports and keeping students healthy). The key findings of the present study were that a 4-min SIT protocol involving three-, 20 s ‘all-out’ effort sprint did not have any acute effect on self-reported mood, alertness, and memory recall in female children and adolescents. These results did not support the authors’ hypothesis that very brief interval exercise enhances student states of learning. Contrary to the present results, previous studies showed that school-based interval training elicited a positive impact on student alertness. A programme called FUNtervals, a six-minute interval exercise that involved four minutes of dynamic, whole-body exercises such as squats, jumping jacks and running on the spot performed at high-intensity showed acute improvements in selective attention in 88 boys and girls aged 9–11 years [29]. In the cited study, the children made fewer errors in the d2 test, an objective measure of one’s selective attention, following the FUNtervals session compared to when they were being assigned to a no-activity break group. This was apparently the only study that examined the effects of HIIT following a brief delay of 11–20 min, as in the present study. It was previously suggested in a meta-analysis that a post-exercise delay of 11–20 min was most likely to elicit positive responses in cognition [2]. Interestingly, a recent study reported that the boost in selective attention in 158 adolescents lasted for an hour after a 16-min HIIT session [38]. The 12- to 16-year-old adolescents in the cited study were instructed to perform 30 s of high-intensity exercise in between rest intervals of 30 s.

Findings in the literature on the effect of school-based interval training on mood in children and adolescents are scarce. The SIT protocol in the present study used only one movement task, that is sprinting, rather than a series of different body movements. By the day of acute assessment, the activity became rather mundane to the participants as commented by one of the PE teachers. This could have dampened their motivation and resulted in the lack of change in their self-reported affect. This view concurred with previous findings from Cooper and his colleagues who reported that 10 × 10 s running sprints, interspersed with 50 s of active recovery had no beneficial immediate effect on self-reported energy, tension, and calmness in adolescents [27]. Participants in the cited study reported a higher level of tiredness following the exercise than when they were seated in the resting trial. It is likely that participants’ mood was in an attenuated state when the mood questionnaire was administered soon after exercise. These results were in contrast with findings reported by another study [28]. They reported that the mood of 21 adolescents improved significantly following an eight- to 10-min HIIT intervention. The reasons for such mixed results are not readily apparent but differences in interval-training protocols, participant cohorts, and the timings of the assessments are plausible explanations.

Few studies have investigated the acute effect of HIIT on memory recall in children or adolescents. Findings in the present study showed negligible effect on memory recall in both primary school children and secondary school adolescents. Similarly, no acute effect on visuo-spatial memory and pictorial memory recall in adolescents were reported in other studies [27,38]. Instead, researchers in the latter cited study showed that selective attention and concentration increased in the second and third hour after the HIIT intervention. It is noteworthy that the HIIT protocol employed in the cited study is four times the duration (16 versus 4 min) of the SIT protocol used in the present study. The interval training protocols used in the cited studies were not identical to that used in the present study. (i.e., work-to-rest ratio; total exercise time). In addition, differing qualitative characteristics of the movement tasks (i.e., cognitive demand and coordinative complexity) may have accounted for the mixed results. It was suggested that activity that requires greater attentional and cognitive resources led to greater extent of improvement in cognition than activities with low cognitive engagement [39,40]. The movement task used in the present study is sprinting which most children and adolescents are quite accustomed to. It also does not require a greater degree of coordination compared to exercises described in other studies. Combining the results of 6 acute studies, the authors of a recent review had found that a single bout of HIIT produced significant yet small to moderate acute effects on executive function and affect in youths [41]. Therefore, whether such brief interval training interventions are useful need to be addressed using different perspectives in different school contexts. For instance, the efficacy of SIT-type programmes that are time-saving and low volume in helping female youths adopt a less sedentary lifestyle outside of school.

The use of a single-item hedonic scale is reported elsewhere and is also used for self-reporting purposes in adolescents [28]. Adolescents in the cited study were asked to complete the hedonic mood scale before and after every HIIT session throughout the period of intervention (a total of 24 times). In contrast to the present findings, adolescents’ mood following HIIT significantly improved by an average of 0.97. Unlike other questionnaires used in the interval training literature, the single-item hedonic mood scale is not established as a validated instrument. Notwithstanding its unestablished validity, the single-item hedonic scale takes less than one minute to answer and is easily comprehensible to children and adolescents.

The total time taken to complete the SIT protocol in the present study is a fraction of the time taken by participants in other studies cited in the literature—i.e., 4 min in present study vs. 10 and 16 min in other studies [27,38]. It is plausible that the exercise dose in the present study was too brief to have any effect on alertness, mood, and memory recall from baseline values (pre-SIT intervention). In the present study, participants reported they were relatively alert and in good mood, and their memory recall scores were not markedly in deficit before the acute SIT intervention. The absence of significant difference is plausibly due to a ceiling effect for improvement [42] since PE sessions were conducted relatively early in the morning of a schooling day where children and adolescents are reasonably rested.

### 4.1. Exercise Adherence and PE Teachers’ Perspective on Infusing SIT in PE Classes

The qualified PE teachers who conducted the study were specially trained by the principal investigator and his team. To motivate participants to perform ‘all-out’ intensity efforts, they were encouraged to match or better the number of sprint-shuttles completed in the previous bouts. Although the proportion of participants who managed to match or better their number of sprint-shuttles is not reported in the present study, a majority had completed all SIT sessions conducted (91.8% and 93.7% of the female children and adolescents, respectively). The exercise adherence in the present study compares well with the exercise adherence rate of 90% among Australian adolescents reported elsewhere [43]. The continued participation even after the third session, when acute assessment of alertness, mood and memory recall were administered, is an indication that SIT is an appealing exercise for female children and adolescents in the context of the present study. Additionally, when PE teachers were asked for their perceptions on the feasibility in incorporating SIT-type activities in PE lessons, ratings provided were very encouraging. Like the present study, several studies had situated the delivery of interval training as an exercise intervention during PE classes, albeit for different purposes [44]. On the balance of discussion, it appears that PE classes could be avenues where SIT-type or HIIT-type activities can feature, given its flexibility in incorporating different forms of dynamic exercise movements as well as its time-saving regimen.

### 4.2. Strengths and Limitations of Study

A unique contribution of the present study was that it involved a cohort entirely of female participants and was one of the largest cross-sectional study that the authors are aware of. Furthermore, the high retention rate and positive ratings from PE teachers are indications of the potential scalability of introducing SIT-type programme in schools. A limitation of the present study was that the acute assessments for mood, alertness, and memory recall were measured on only one occasion and specifically within the period of 11–20 min (i.e., about 15 min) after the SIT. It is indeterminate in the present study if the prescribed SIT protocol has any abbreviated or transient effect on the aforementioned factors that affects learning outside of the 11- to 20-min window. In addition, as this study was conducted only on female participants, the effect of SIT on male participants remains to be examined.

### 4.3. Future Research Directions

Future research could explore different permutations on the work-to-rest ratio of the SIT protocol and examine its acute effect on mood, alertness, and memory recall of children and adolescents in a school-based setting. Interval-training could be embedded during the latter part of a school day when mood, alertness and memory of students are on the wane. These brief exercise breaks should be low in volume so that it does not take up much of class time and they should be curated with participant enjoyment in mind.

## 5. Conclusions

A 4-min SIT involving three, 20 s ‘all-out’ effort intensity sprints had no acute effect on improving mood, alertness and memory in female children and adolescents. The high exercise adherence rate and encouraging ratings by PE teachers are suggestive of the potential scalability of incorporating SIT into PE programme in schools. There is a need for more school-based research to explore the acute effect of different SIT permutations in the context of each school.

## Figures and Tables

**Figure 1 jfmk-05-00092-f001:**
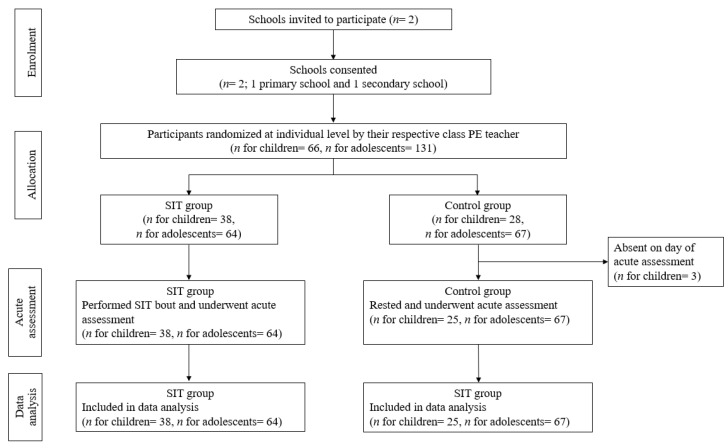
Flow diagram through each stage of the parallel, randomized control study design.

**Table 1 jfmk-05-00092-t001:** Mean ± SD of alertness, mood, and memory recall of female primary school children.

	SIT Group (*n* = 38)			Control Group (*n* = 25)				Repeated Measures ANOVA		
	Pre-exercise	11–20 min post-exercise	Change	Pre-intervention	11–20 min post-condition	Change	Effect size of changes between groups, *η_p_*^2^	Time (*p*-value)	Group (*p*-value)	Time * group (*p*-value)
Alertness	6.9 ± 2.1	6.8 ± 2.5	−0.1	6.9 ± 1.7	6.6 ± 1.1	−0.3	0.001	0.353	0.884	0.632
Mood	6.2 ± 2.4	6.1 ± 2.0	−0.1	6.2 ± 2.1	6.6 ± 1.7	0.4	0.006	0.671	0.545	0.360
Memory recall	6.5 ± 1.8	7.6 ± 2.1	1.1	7.0 ± 2.0	7.3 ± 1.7	0.3	0.001	0.018	0.927	0.177

* Interaction between time of assessment and experimental group.

**Table 2 jfmk-05-00092-t002:** Mean ± SD of alertness, mood, and memory recall of female secondary school adolescents.

	SIT Group (*n* = 64)			Control Group (*n* = 67)				Repeated Measures ANOVA		
	Pre-exercise	11–20 min post-exercise	Change	Pre-intervention	11–20 min post-condition	Change	Effect size of changes between groups, *η_p_*^2^	Time (*p*-value)	Group (*p*-value)	Time * group (*p*-value)
Alertness	6.1 ± 1.5	6.1 ± 1.5	0	6.0 ± 2.1	5.9 ± 1.9	−0.1	0.003	0.687	0.556	0.534
Mood	6.1 ± 1.5	6.0 ± 1.6	−0.1	5.7 ± 2.2	5.7 ± 2.0	0	0.11	0.745	0.237	0.442
Memory recall	10.5 ± 2.2	9.9 ± 2.2	−0.6	10.8 ± 2.2	10.0 ± 2.3	−0.8	0.001	0.001	0.825	0.707

* Interaction between time of assessment and experimental group.

## Data Availability

Data used in the manuscript will be deposited to the data repository of the corresponding author’s institution. Permission from the corresponding author before accessing the data is required.

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
