# Peer review of "Four Minutes of Sprint Interval Training Had No Acute Effect on Improving Alertness, Mood, and Memory of Female Primary School Children and Secondary School Adolescents: A Randomized Controlled Trial"

_jfmk, 2020, doi:10.3390/jfmk5040092_

Round 1

Reviewer 1 Report

This is an interesting article, well designed and well written.

I would suggest to better discuss your results and try to interpret these unexpected results, also because in the introduction you already state that the aerobic exercise has demonstrated as being effective and the interval training seems potentially effective. 

Author Response

Comment by reviewers

Authors’ responses

Reviewer no. 1:

I would suggest to better discuss your results and try to interpret these unexpected results, also because in the introduction you already state that the aerobic exercise has demonstrated as being effective and the interval training seems potentially effective.

Thank you, reviewer, for your feedback. I have explained the unexpected results in greater details and supported the explanations with new references. Please refer to line 239-243, and line 262-270.   

Reviewer no. 2:

On line 181 you wrote that you used the partial eta squared (squared but not square) to measure the effect size. Therefore, I advise you to replace 'η2' with 'ηp2' throughout the manuscript, including the tables

Thank you, reviewer, for pointing out what the correct symbol for partial eta squared should be. We have replaced the symbol throughout the manuscript as well as in the tables.

Reviewer no. 2:

Remove “acute effect, mood, alertness and memory” and, if necessary, add other relevant key words. I would add, for example, "youth learning" instead "learning in children and adolescents".

Thank you, reviewer, for your suggestions. I have removed the key words according to your suggestions and added the following new key words: learning in youths; acute exercise; school; attention.

Reviewer no. 2:

Be careful to indicate the bibliographic references: you do not write 3-4 but 3,4, or 7,8 and so on. For two successive references the comma is put, if there are three or more successive references then the hyphen is used. Check the entire manuscript.

Thank you, reviewer, for your feedback. I have made the necessary corrections for successive references and checked through the entire manuscript.

Reviewer no. 2:

Line 92: Replace “Methodology” with “Materials and Methods”

I have amended accordingly.

Reviewer no. 2:

Lines 188-191: the statistical analysis used is not clear to me. I think you used MANOVA here and so you must point it out

Thank you, reviewer, for your comment. Yes, the results reported inline 188-191 were derived from MANOVA. I have included the use of MANOVA in line 189.

Reviewer no. 2:

Table 1 e Table 2: I cannot understand what the partial eta squared  in this table refers to. Does it refer to the comparison between the pre-post-intervention changes in the two groups? In any case you must indicate it in the caption. It is important that the tables are independent of the text. The reader must understand without having to review parts of the manuscript

Yes, the partial eta squared in Table 1 and Table 2 refers to the difference in pre-to-posttest changes between the two groups (SIT group and CON group). I have indicated more clearly in the revised manuscript. Thank you.

Reviewer 2 Report

General comments

The authors presented an interesting study that investigated  whether a 4-min sprint interval training protocol had an acute effect on improving alertness, mood, and memory recall in female students; also, a secondary purpose was to  gain perspectives  from  PE  teachers  about  the  feasibility of infusing training protocol into PE classes. The results showed that the protocol did not have an acute main effect on improving alertness,  mood  and,  memory  recall  in  female  children and  adolescents. Evaluations of PE teachers have suggested the potential scalability of integrating the protocol into the PE curriculum in schools.  

In my opinion, the manuscript is well written and has shown preliminary results which will need to be studied with further research; however, it needs a review to make it suitable for publication in JFMK. To this end, I have some suggestions for authors to improve the manuscript.

Specific comments

Overall manuscript

-On line 181 you wrote that you used the partial eta squared (squared but not square) to measure the effect size. Therefore, I advise you to replace 'η2' with 'ηp2' throughout the manuscript, including the tables.

Keywords

-To optimize the search of the manuscript on the search engines, insert different keywords from those present in the title.

Remove “acute effect, mood, alertness and memory” and, if necessary, add other relevant key words. I would add, for example, "youth learning" instead "learning in children and adolescents". Of course, these are just suggestions. It is important to add words other than those present in the title.

Introduction

-Be careful to indicate the bibliographic references: you do not write 3-4 but 3,4, or 7,8 and so on. For two successive references the comma is put, if there are three or more successive references then the hyphen is used. Check the entire manuscript.

Materials & Methods

-Line 92: Replace “Methodology” with “Materials and Methods”

-Lines 188-191: the statistical analysis used is not clear to me. I think you used MANOVA here and so you must point it out.

-Table 1 e Table 2: I cannot understand what the partial eta squared  in this table refers to. Does it refer to the comparison between the pre-post-intervention changes in the two groups? In any case you must indicate it in the caption. It is important that the tables are independent of the text. The reader must understand without having to review parts of the manuscript.

Author Response

(The authors gave the same response as above.)

Round 2

Reviewer 2 Report

Thanks for your replies.